



# Global distribution of photosynthetically available radiation on the seafloor

Jean-Pierre Gattuso[1, 2], Bernard Gentili[1], David Antoine[1, 3], and David Doxaran[1]

[1]Sorbonne Université, CNRS, Laboratoire d'Océanographie de Villefranche, 181 chemin du Lazaret, F-06230 Villefranche-sur-mer, France
[2]Institute for Sustainable Development and International Relations, Sciences Po, 27 rue Saint Guillaume, F-75007 Paris, France
[3]Remote Sensing and Satellite Research Group, Curtin University, Perth, Western Australia, 6845, Australia

**Correspondence:** J.-P. Gattuso (gattuso@obs-vlfr.fr)

**Abstract.**

A 21-year (1998–2018) continuous monthly data set of the global distribution of light (photosynthetically available radiation; $PAR$) reaching the seabed is presented. It uses ocean colour and bathymetric data to estimate benthic irradiance, offering critical improvements on a previous data set (Gattuso et al., 2006). The time series is 4 times longer (21 vs 5 years), the spatial resolution is better (pixel size of 4.6 vs 9.3 km at the equator) and the bathymetric resolution is also better (pixel size of 0.46 vs 3.7 km at the equator). The paper describes the theoretical and methodological bases and data processing. This new product is used to estimate the surface area of the sea floor where (1) light does not limit the distribution of photosynthetic benthic organisms and (2) net community production is positive. The complete data set is provided as 14 netCDF files available on PANGAEA (Gentili and Gattuso, 2020, https://doi.pangaea.de/10.1594/PANGAEA.910898). The R package CoastalLight, available on Github (https://github.com/jpgattuso/CoastalLight.git), allows (1) to download geographical and optical data from PANGAEA and (2) to calculate the surface area that receives more than a given threshold of irradiance in three regions (non polar, Arctic and Antarctic). Such surface areas can also be calculated for any sub-region after downloading data from a remotely and freely accessible server.

## 1 Introduction

Light is a key ocean variable. It shapes the composition of benthic and pelagic communities by controlling the three-dimensional distribution of primary producers, the lowest levels of the wood webs. Light also plays a major role in the global carbon cycle by controlling primary production, the main source of new organic carbon in the ocean (Assis et al., 2018). In the marine environment, sunlight is rapidly absorbed by the water column and primary production is restricted to the shallow photic zone above 200 m depth (except for localized chemo-autotrophic communities). Marine diazotrophs, which fix dinitrogen into organic forms, are also light-dependent. Furthermore, many marine ecosystem engineers require light because they are either plants (mangrove, saltmarshes, seagrass, coralline algae) or animals living in symbiosis with endosymbiotic algae (e.g., some molluscs and zooxanthellate, reef-building corals).





Until the late 1970s, most water transparency measurements were performed using Secchi disks (Tyler, 1968) and several formulations became available to convert Secchi disk readings to attenuation coefficients (e.g., Weinberg, 1976). Remote sensing observations of ocean colour showed great promise as early as 1978, when the Coastal Zone Color Scanner (CZCS) was launched. It was followed by several other instruments on-board satellites. Ocean colour measurements of the Sea-Viewing

Wide Field-of-View Sensor (SeaWiFS), launched in 1997, are used to derive the concentration of chlorophyll-a ($C_{sat}$) and the mean attenuation coefficient for $PAR$ ($K_{PAR}$). Until 2006, most attention was focused on the light field in the water column to derive open-ocean primary production (e.g., Antoine et al., 1996). However, primary production also occurs in the coastal ocean when enough light reaches the sea floor. For example, on coral reefs, benthic primary production can represent 90% of the total primary production (Delesalle et al., 1993). Primary production in coastal vegetated habitats such as mangroves,

seagrass beds and tidal marshes, the so-called 'blue carbon ecosystems', has received considerable interest in the past 10 years. because of their disproportionately large contribution to global carbon sequestration (Macreadie, 2019). It has been recently suggested that benthic macroalgae also contribute to global carbon burial (Krause-Jensen et al., 2018).

Gattuso et al. (2006) used SeaWiFS data collected between 1998 and 2003 to estimate, for the first time at a nearly global scale, the irradiance reaching the bottom of the coastal ocean. They provided cumulative functions to estimate the percentage

of the surface ($S$) of the coastal zone receiving more than a given irradiance. These data were used to investigate the extent of macroalgae (Krause-Jensen and Duarte, 2016), restoration of seagrass ecosystems (Eriander, 2017), role of vegetated coastal habitats in the ocean carbon budget (Duarte, 2017), macroalgal subsidies supporting benthic invertebrates (Filbee-Dexter and Scheibling, 2015), global continental shelf denitrification (Eyre et al., 2013), and benthic primary production in the Arctic Ocean (Attard et al., 2016; Glud et al.).

More recently, Assis et al. (2018) provided a data layer for benthic irradiance for species distribution modelling as part of the Bio-ORACLE set of GIS rasters. This data set is based on $K_{d,490}$ in contrast to Gattuso et al. (2006) who used the more appropriate $K_{PAR}$ to estimate bottom $PAR$ ($PAR_B$). This is particularly important in coastal regions where there is no unique relationship between $K_{d,490}$ and $K_{PAR}$ due to large differences in the concentration and composition of non-algal coloured substances.

Since these first efforts, new products have become available which can improve estimates of the global distribution of benthic irradiance. These include a much longer time series of ocean colour (21 vs 5 years) with an improved spatial resolution (4.6 vs 9.3 km at the equator). Bathymetric data have also considerably improved since 2006 (0.46 vs 3.7 km at the equator). Here we make use of these new products to provide a global distribution of photosynthetically available radiation reaching the seafloor.

## 30  2 Methods

The characteristics of the products used by Gattuso et al. (2006) and in the present study are compared in Table 1 .





**Table 1.** Main characteristics of the products used by Gattuso et al. (2006) and in the present study.

|  | Gattuso et al. (2006) | Present study |
|---|---|---|
| Satellite | Jan 1998   SeaWiFS   Dec 2003 | Jan 1998   SeaWiFS   Dec 2010 <br> May 2002   MERIS   Apr 2012 <br> Jul 2002   MODIS   Dec 2018 <br> Feb 2012   VIIRS   Dec 2018 |
| Coverage | 1998 to 2003 | 1998 to 2018 |
| Sat. resolution | $\approx 1/12° = 9.3$ km at equator | $\approx 1/24° = 4.6$ km at equator |
| Bathymetry | ETOPO 2 min | GEBCO 15 sec |
|  | 3.7 km at equator | 0.46 km at equator |
| Data | $PAR$, $C_{sat}$, $nLw(555)$, $K_d$ from $C_{sat}$ | $PAR$, $K_{PAR}$, $C_{sat}$, $R_{rs}(555)$ |

## 2.1 Remote sensing data

Monthly Level-3 data of $PAR$ (mol photons m$^{-2}$ d$^{-1}$), $K_{PAR}$ (m$^{-1}$), concentration of chlorophyll-a ($C_{sat}$, mg m$^{-3}$), and remote sensing reflectance at 555 nm ($R_{rs}(555)$, sr$^{-1}$) from the four satellite-borne sensors SeaWiFS, Moderate Resolution Imaging Spectroradiometer (MODIS), MEdium Resolution Imaging Spectrometer (MERIS) and Visible Infrared Imaging Radiometer Suite (VIIRS) were obtained from the GlobColour project (http://www.globcolour.info). The resolution is 1/24°. Together, the 252 monthly images downloaded (a level-3 image contains values of a product on a regular longitude-latitude grid) cover the period 1998 to 2018.

## 2.2 Bathymetry and coastline

Depths were estimated from the 2019 General Bathymetric Chart of the Oceans (GEBCO; https://www.gebco.net) gridded bathymetry data (1/240° resolution). The coastal zone (0 to 200 m) was determined using a land mask and coastline (Global Self-consistent, Hierarchical, High-resolution Geography, GSHHG) as implemented in the Generic Mapping Tools (GMT; Wessel et al., 2013). The full resolution was used. The Arctic, Antarctic, and non polar regions represent, respectively, 24.1, 0.6, and 75.3% of the surface area of the coastal zone.

## 2.3 Case 1 versus Case 2 waters

It is beyond the scope of this paper to review the criteria used to eliminate dubious data when generating a Level-3 ocean colour composite, except for discriminating the water type as being either Case 1 or Case 2 (Morel and Prieur, 1977). In Case 1 waters, where phytoplankton and associated degradation products are the main contributors to light attenuation (but see Claustre and Maritorena, 2003), $K_{PAR}$ can be modelled as a function of the concentration of chlorophyll-a, itself derived from reflectance values. The situation is, however, not as straightforward in Case 2 coastal waters where light attenuation by coloured dissolved organic matter and suspended particles other than phytoplankton can be significant and not correlated to the chlorophyll-a concentration. The discrimination between these two types is performed at the Level-2 in the processing, yet it is





not considered when generating the Level-3 composites (B. Franz, personal communication, September 2019). Therefore, the average chlorophyll-a concentration $C_{sat}$ in a given bin of a Level-3 composite may have been computed over any proportion of Case 1 and Case 2 waters.

The accuracy of $C_{sat}$ in Case 1 waters is claimed to be $\pm 30\%$ whereas it is unknown in Case 2 waters. It is therefore not possible to estimate the accuracy of the chlorophyll product in coastal areas and, in turn, the accuracy of the diffuse attenuation coefficient. The determination of the water type could not be performed with specific algorithms for each water type since no universal algorithm exists for Case 2 waters. It was was carried out *a posteriori* based on the average $C_{sat}$ and $R_{rs}(555)$. This determination provides an indication of bins that likely belong to the Case 2 water category when, on average, the individual pixels accumulated in the bins were predominantly of the Case 2 type.

The identification of turbid Case 2 waters has been performed as in Morel and Bélanger (2006) by comparing the water reflectance at 555 nm ($R(555)$) to the maximum value it should have in Case 1 waters and for the same chlorophyll concentration ($R_{lim}(555)$). Note that the water type was set to Case 1 for any pixel where $C_{sat} < 0.2$ mg m$^{-3}$, because the algorithm is occasionally subject to falsely classify low-chlorophyll waters as Case 2 Morel and Bélanger (2006). Turbid Case 2 waters are those for which $R(555) > R_{lim}(555)$. To perform this test, $R_{rs}(555)$ was converted to $R(555)$ as follows (Morel and Gentili, 1996):

$$R(555) = R_{rs}(555) \times Q_0(555)/\Re_0 \tag{1}$$

where $Q_0(555)$ is the chlorophyll-dependent $Q$-factor (sr), i.e., the ratio of the upward irradiance to the upwelling radiance (Morel et al., 2002), and $\Re_0$ is a term that merges all reflection and refraction effects at the air-sea interface (0.529). Since $R_{rs}(555)$ is fully normalized (Morel and Gentili, 1996), its dependence on the viewing angle and the sun zenith angle are removed so that both $Q$ and $\Re$ are taken for a nadir view and a sun at zenith (hence the "0" subscript).

### 2.4 Benthic irradiance

The diffuse attenuation coefficient for the downward irradiance $K_d(\lambda_0)$ for a given wavelength $\lambda_0$ describes the exponential attenuation of irradiance with depth in the water column. It determines the amount of radiation reaching any given depth:

$$K_d(\lambda_0, z) = \frac{-\partial \ln(E_d(\lambda_0, z))}{\partial z} \tag{2}$$

The spectral composition of the radiation is not considered in this work and only its integral value between 400 and 700 nm is used (i.e., the photosynthetically available radiation, $PAR$). The attenuation coefficient for $PAR$ is therefore:

$$K_{PAR}(z) = \frac{-d \ln(PAR(z))}{dz} \tag{3}$$





The average value $K_{PAR}$ of $K_{PAR}(z)$ over the euphotic zone, approximated as the depth where $PAR$ is reduced to 1% of its value just beneath the sea surface, is computed from the corresponding chlorophyll concentration for Case 1 waters $C_{sat}$ and $K_d(490)$ using the following equations (Morel et al., 2007; ACRI-ST GlobColour Team, 2017) :

$$K_d(490) = 0.0166 + 0.08349 \times C_{sat}^{0.63303} \tag{4}$$

$$K_{PAR} = 0.0665 + 0.874 \times K_d(490) - 0.00121/K_d(490) \tag{5}$$

The bottom irradiance is then calculated:

$$PAR_B = \exp(-K_{PAR} \times z) \tag{6}$$

- for the Non-Polar region all months are taken into account, so we have 21 years $\times$ 12 months = 252 values by pixel at most

- for the Arctic region months 6-10 (June to October) are taken into account, so we have 21 years $\times$ 5 months = 105 values by pixel at most

- for the Antarctic region months 1-3 and 11-12 (January to March and November-December) are taken into account, so we have 21 years $\times$ 5 months = 105 values by pixel at most

- in fine, we have 252 monthly $PAR_B$ images for the non polar region and 105 for the Arctic and Antarctic regions.

The product delivered comprises longitude, latitude, depth, area, $PAR$, $K_{PAR}$, $PAR_B$ for each coastal pixel. $PAR$, $K_{PAR}$ and $PAR_B$ are monthly climatologies or a climatology over the entire time series (see Section 4). The calculation of surface area receiving $PAR_B$ above a certain threshold does not use these climatologies.

## 2.5   Surface area receiving light above a certain threshold

Calculations of surface area receiving $PAR_B$ above a certain threshold are made in two steps. First a $\mathcal{P}$-function is calculated
20 with the available pixels; then the area is calculated as the product of the $\mathcal{P}$-function by the surface of the coastal zone (0-200 m).

### 2.5.1   The three main regions

A region is defined here by an interval of latitude at the surface of the Earth. Polar regions are more frequently observed by satellites, yet polar night and cloudiness end up with data not being available several months a year. So three regions have been
25 defined:

- the "non polar" region $[60°S; 60°N]$ , where data are always available,





- the "Arctic" region $[60°\mathrm{N}; 90°\mathrm{N}]$, where data are available during the months of June, July, August, September, and October,

- the "Antarctic" region $[90°\mathrm{S}; 60°\mathrm{S}]$, where data are available during the months of January, February, March, November, and December.

## 2.5.2 $\mathcal{P}$-functions

**Definition of a $\mathcal{P}$-function for a monthly $PAR_B$ image of a region**

- let $I$ be the monthly image (values of $PAR_B$ on the floor of the coastal zone of the region)

- let $S_{a,I}$ be the available surface, i.e. the total surface of pixels for which an irradiance value is available (varying every month);

- let $E$ a value of irradiance (expressed in $\mathrm{mol\ photons\ m^{-2}\ d^{-1}}$);

- let $s_I(E)$ the total surface of pixels collecting irradiance greater than $E$;

- the $\mathcal{P}_I$-function if defined as $\mathcal{P}_I(E) = 100 s_I(E)/S_{a,I}$

**Definition of a climatologic $\mathcal{P}$-function**

Our purpose is now to define a $\mathcal{P}$-function for a set of monthly values $\mathcal{I} = \{I_i, i = 1\ldots n\}$. Giving a value of irradiance $E$, it is defined as :

$$\mathcal{P}_\mathcal{I}(E) = 100 \sum_{i=1}^{n} s_{I_i}(E) / \sum_{i=1}^{n} S_{a,I_i} \tag{7}$$

**Climatologic monthly $\mathcal{P}$-function**

In this case, the 21 data sets available for a given month through the entire time-series (1998 to 2018) are selected to calculate the $\mathcal{P}$-function according to equation 7. So we have :

- 12 climatologic monthly $\mathcal{P}$-functions for the Non-Polar region,

- 5 climatologic monthly $\mathcal{P}$-functions for the Arctic region,

- 5 climatologic monthly $\mathcal{P}$-functions for the Antarctic region.

**Climatologic global $\mathcal{P}$-function**

$\mathcal{P}_g$ is obtained, using all data sets (252 for Non-Polar and 105 for Arctic and Antarctic) in equation 7.





**$\mathcal{P}$-functions for a subregion**

Sub-region may be defined within one of the three main regions. In this case data sets are clipped according to the sub-region's boundaries, and the months used are those of the main region. Calculation is identical to that described above for the climatological global $\mathcal{P}$-function (section 2.5.2). The R package *CoastalLight* (see Section 4) can be used to calculate a $\mathcal{P}$

5 function for a subregion.

**2.5.3 Surface areas**

Let $\mathcal{P}$ be the $\mathcal{P}$-function of the zone and $S_{geo}$ its area : the area receiving irradiance above a threshold $E$ is :

$$s(E) = S_{geo} \frac{\mathcal{P}(E)}{100} \tag{8}$$

**3 Results and discussion**

10 The present study essentially confirms the bathymetric data reported in our earlier study (Gattuso et al., 2006) but shows substantial differences on the optical data.

**3.1 Surface area and depth of sub-regions of the ocean**

**Table 2.** Surface area ($S$) of coastal waters (depth $<$ 200 m) of different optical characteristics. Calculations were performed on monthly products. Values reported by Gattuso et al. (2006) are shown in parentheses for comparative purposes. Gattuso et al. (2006) did not report data for the Antarctic.

|  | Arctic | | Antarctic | | Non Polar | |
|---|---|---|---|---|---|---|
|  | $S(10^6 km^2)$ | $S(\%)$ | $S(10^6 km^2)$ | $S(\%)$ | $S(10^6 km^2)$ | $S(\%)$ |
| Coastal Zone | 6.1 (6.13) | 100 (100) | 0.146 | 100 | 19.1 (18.8) | 100 (100) |
| Case 1 | 2.37 (1.6) | 38.8 (26.2) | 0.029 | 20.1 | 11.3 (8.47) | 59.2 (45) |
| Case 2 | 0.72 (0.81) | 11.8 (13.2) | 0.022 | 14.8 | 4.62 (6.76) | 24.2 (35.9) |
| Case 1 and Case 2 | 3.08 (2.41) | 50.5 (39.40) | 0.051 | 34.9 | 15.9 (15.23) | 83.4 (80.9) |

The area and depth of the three regions measured with the most recent GEBCO bathymetry are very similar to those obtained with the coarser ETOPO2 data set used by Gattuso et al. (2006) (Table 2). The surface area of the ocean with less than 200

15 m depth is 25.3 $10^6$ km$^2$. Three geographical areas are considered: the Arctic (60 to 90°N), non polar (60°N to 60°S), and Antarctic (60 to 90°S) regions, respectively covering 24.1, 75.5 and 0.6% of the global coastal zone. The average depth of the coastal zone is almost twice as large in the Antarctic than in the Arctic and non polar regions (137 vs 77 and 71 m).

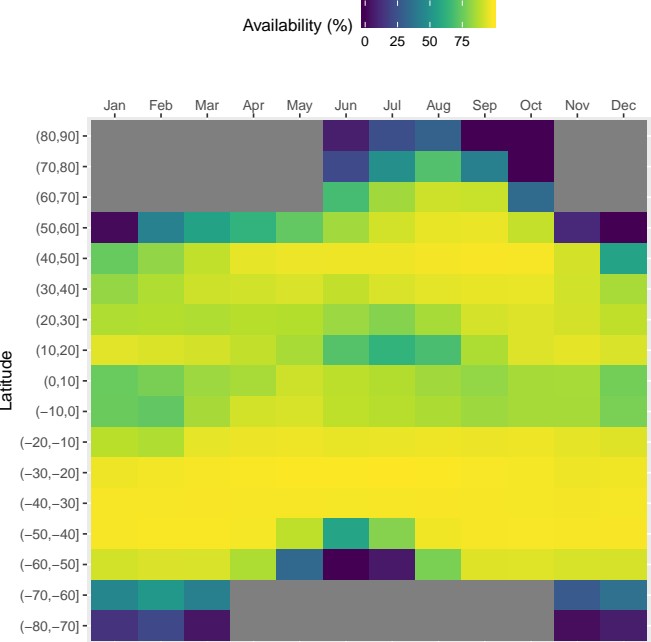

**Figure 1.** Mean monthly availability of remote sensing data over the entire 21 years time-series expressed as the percent of the surface of the coastal zone in each latitudinal band.

## 3.2 Availability of ocean colour data and seawater types

The availability of monthly ocean colour data is highly variable depending on the latitude and month of the year (Fig. 1). It is highest in non polar regions where, on average, data are available in 83% (range: 62-96%) of the pixels in monthly data sets. There is light for only 5 summer months of the year in the Arctic (June to October) and Antarctic (November to March).

During these periods, data availability is higher in mid-summer than in early- and late summer (Fig. 1). Data availability also decreases as one gets closer to the poles. On average, data are available for 51 and 35% of the summer data sets in the Arctic and Antarctic regions (ranges: 6-89% and 11-58%, respectively; 3). It is higher in the present study which used multiple sensors than in a previous study that only used SeaWiFS data (Gattuso et al., 2006). Several factors contribute to the lower availability of data in polar regions: pixels are contaminated by sea ice and flagged accordingly, high occurrence of cloudy days and low

incidence of the sun.

The coverage of the Arctic has improved with about 20% more pixels with available data (Table 3). Case 1 and Case 2 waters are approximately equally distributed in the Antarctic region (Table 2). In contrast, the distribution of Case 1 and Case 2 waters in the non polar region, with a clear dominance of Case 1 over Case 2 waters (70 vs 30%) in the present study whereas it was more even in Gattuso et al. (2006, 55 vs 45%). This difference may be due to the different approaches used to

15 differentiate Case 1 and Case 2 waters. The present study used the remote sensing reflectance at 555 nm ($R_{rs}(555)$) provided

by the GlobColour project whereas it was roughly estimated from the normalized water-leaving radiance in the previous study
(Eq. 1 in  Gattuso et al., 2006). The quality of the results should therefore have improved. In any case, the usefulness of this
distinction is relatively limited because the light penetration through the water column is calculated in the same way in the two
cases. The distribution of water quality is however useful to estimate the reliability of the bottom irradiance which is much
better in Case 1 waters than in Case 2 waters. The average depth of the missing pixels is similar to that of the available pixels
in the Arctic and Antarctic regions (Table 3). However, it is sometimes lower in the non polar region. The lowest values occur
when the amount of available pixels is the largest (data not shown), suggesting that the missing pixels are preferentially located
close to the coastline.

### 3.3   Bottom irradiance

The distribution of $PAR_B$ has changed in the present study compared to the previous one of Gattuso et al. (2006), with less
irradiance values above 0.2 mol photons m$^{-2}$ d$^{-1}$ and more irradiance values around 0.1 mol photons m$^{-2}$ d$^{-1}$ in 2019 than
in 2006 (Fig. 2). Consequently, the surface area receiving irradiance above a certain threshold also declined (Fig. 3).

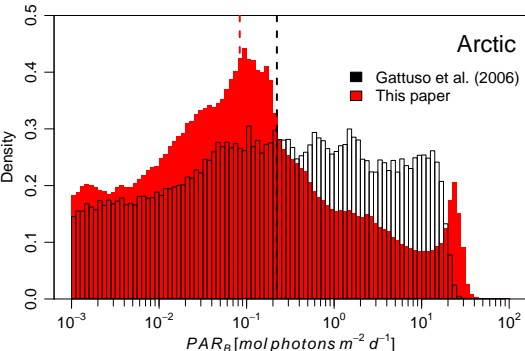 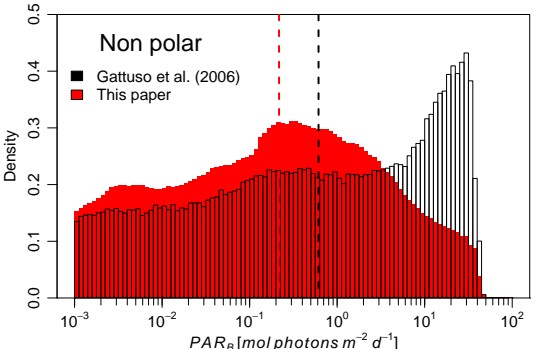

**Figure 2.** Distribution of $PAR_B$ in the present study (2019) and in Gattuso et al. (2006). The vertical dashed lines represent the median
values in 2006 (black) and present (red) studies.

The surface area of the sea floor receiving an irradiance larger than a threshold value is lower than with the previous estimate
of Gattuso et al. (2006)(Fig. 3). Differences are low below an irradiance threshold of 0.2 mol photons m$^{-2}$ d$^{-1}$: 3 to 16%
lower, respectively in the non polar and Arctic regions. However, differences are as high as 26 and 56%, respectively in the non
polar and Arctic regions for irradiance thresholds ranging between 10 and 50 mol photons m$^{-2}$ d$^{-1}$. Such differences can be
due to several causes.

The present study and Gattuso et al. (2006) used different approaches. In the 2006 study, a p-function was derived for each
month and then monthly means calculated, implicitly giving the same weight to each month, irrespective of the number of
pixels with available data. In the present study, each month has a weight proportional to the surface area for which data are
available, hence providing better estimates. Second, there are more data available in the data set compiled in the present paper,

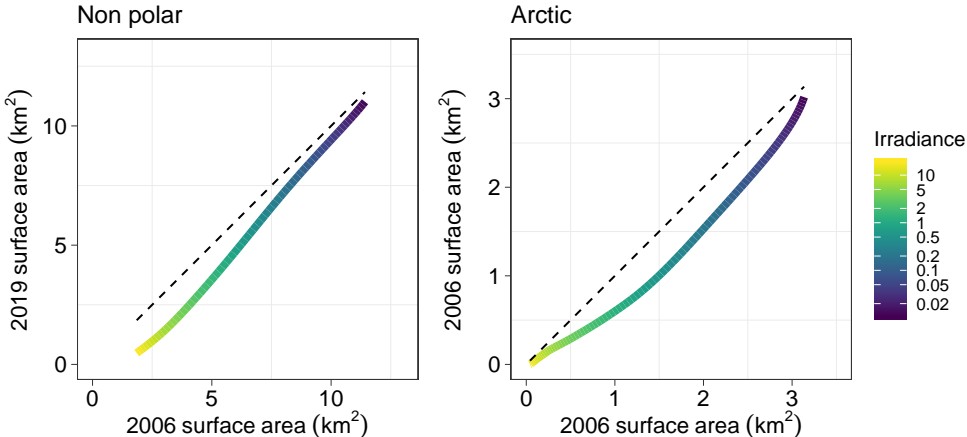

**Figure 3.** Comparison of the surface area of the sea floor receiving an irradiance larger than a threshold value ranging from 0.01 to 20 mol photons $m^{-2}$ $d^{-1}$ calculated in the present paper compared with the surface area reported in 2006 by Gattuso et al. (2006). The dotted line is the 1:1 relationship.

especially in the Arctic. Third, Gattuso et al. (2006) fitted polynomial functions on the relationship between irradiance and the cumulative surface area of the sea floor receiving irradiance above a prescribed threshold. These functions only provide rough estimates and are not used in the present study. They are shown for comparative purposes in Fig. A1. The R package CoastalLight has been developed in the present study to provide more accurate estimates (Section 4) calculated from the underlying data, that is the number of pixels and their size.

These changes in approach together with the different data sets used for the optical and bathymetric data have led to significant changes in three factors that affect bottom $PAR$ (Fig. 4, Table 4). Two of them contribute to a decline of bottom $PAR$: (1) a change in the depth distribution leading to an increase in the median depth (39 vs 31 m) and (2) the distribution of $K_{PAR}$ moved towards larger values in 2019. Also, (3) surface $PAR$, which controls $PAR_B$, tends to be higher in the present study than in the previous one. We do not have any independent confirmation of such an increase in surface $PAR$ globally. The change could be real but could also result from successive reprocessing of the individual sensor archives that made up the GlobColour products that have occurred since 2006. These reprocessing indeed include updates of calibration coefficients and possible refinements of algorithms. The combined effects of the first two causes are larger that the effect of the third one, explaining why bottom $PAR$ is overall smaller in the present study than in the previous one (Gattuso et al., 2006).



**Figure 4.** Distribution of depth, $K_{PAR}$ and PAR in the present study and in Gattuso et al. (2006). The vertical dashed lines represent the median values in 2006 (black) and present (red) studies.

## 3.4 Implications for the distribution of photosynthetic organisms and communities

The differences in $PAR_B$ between the 2006 study and the present one have implications on the potential surface areas receiving enough irradiance to sustain growth of photosynthetic organisms and communities (Table 5). Surface areas are 4 to 47% lower in the present study depending on the region and organism or community considered. As shown in Fig. 3, in the non polar



region the highest the irradiance threshold, the largest the difference. Hence, the differences are generally reasonable (less than 15%) for organisms but higher (up to 47%) for communities which have higher light requirements to maintain positive rates of net primary production. Differences between the 2006 estimates and the present ones are generally larger in the Arctic than in the non polar region for organisms and fairly similar for communities.

## 3.5  Analysis of time series

Long-term changes in the optical characteristics have recently been described. For example, using SeaWiFS monthly global ocean transparency data over Sep. 1997 to Nov. 2010, He et al. (2017) described a rapid decrease in global mean ocean transparency at a rate of -0.85 m yr$^{-1}$ between 1997 and 1999, followed by a small increase with a rate of 0.04 m yr$^{-1}$ over 2000–2010.

In the Arctic coastal zone, significant climate change effects have been observed over the last two decades including enhanced melting of sea-ice during the summer period, permafrost thaw and increase of river discharge into the Arctic Ocean. Time-series of ocean color satellite data have been successfully used to confirm these changes and quantify an increase of up to 40% in the concentrations of both dissolved and particulate terrestrial substances in Arctic coastal waters (Doxaran et al., 2015, Matusoka, pers. comm.). In non polar regions, satellite observations did not reveal such significant temporal trend (e.g., Loisel et al., 2014) but often highlighted how human-induced activities impact on the discharge of big rivers and its consequences on the turbidity of surrounding coastal waters (Feng et al., 2014, e.g., ).

With a time series 21 years long, it is tempting to investigate whether long term changes in $PAR_B$ can be identified. Fig. 5 shows the percent surface area of the coastal zone of the non polar region receiving 2 mol photons m$^{-2}$ d$^{-1}$ or more. There is a highly significant trend with an increase in percent surface area of 0.1% ± 0.02 per year (± 99% confidence interval). However, separate regression analyses show data shifts occur between the three time periods when the same ocean colour sensors were in operation. The trends are therefore highly variable during specific time periods corresponding to various sets of ocean colour sensors. We conclude that no long-term trend in $PAR_B$ can be identified in this data set.

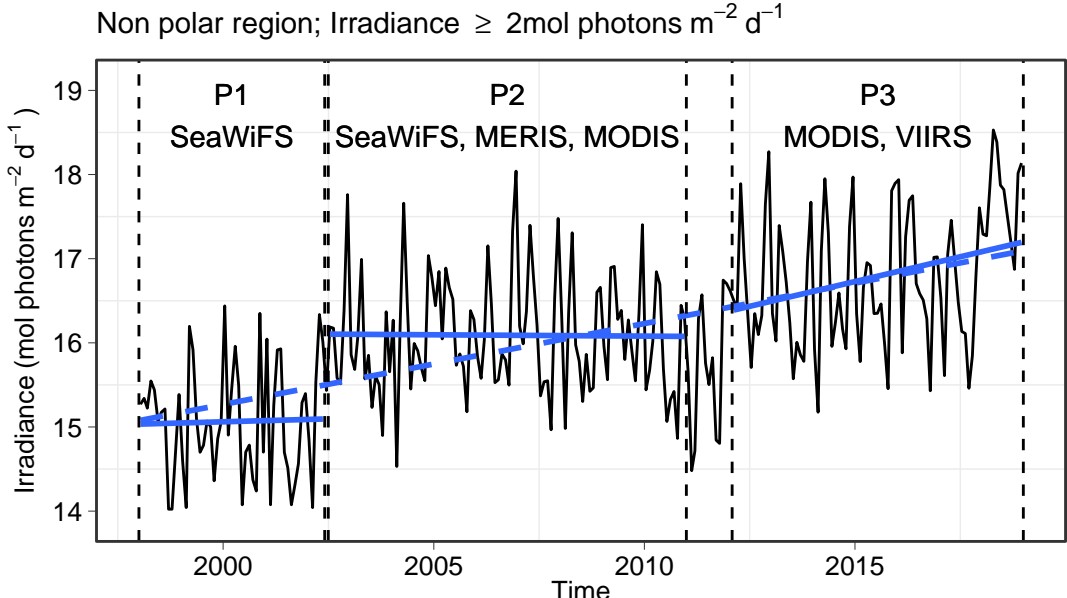

**Figure 5.** Time series of the surface area (%) of the coastal non polar region receiving more than $2\ \mathrm{mol\ photons\ m^{-2}\ d^{-1}}$. The linear regression over 1999-2018 is shown as a dashed line while the result of separate linear regressions for the three time periods with the same set of ocean colour sensors is shown as a solid line.





**Table 3.** Surface area and average depth of the various pixel classes. Calculations were performed on monthly data sets for the periods indicated. Values reported by Gattuso et al. (2006) are shown in parentheses for comparative purposes. Gattuso et al. (2006) did not report data for the Antarctic. $Z_{1\%}$ is the depth at which benthic irradiance equals 1% of surface irradiance. Available pixels are the pixels for which $PAR$, $K_{PAR}$, $C_{sat}$, and $R_{rs}(555)$ are available for analysis.

| | Arctic (June-October) | | | Antarctic (November-March) | | | Non Polar (January-December) | | |
| --- | --- | --- | --- | --- | --- | --- | --- | --- | --- |
| | Min | Max | Mean | Min | Max | Mean | Min | Max | Mean |
| Available/Total number of pixels | 0.059 (0.20) | 0.89 (0.60) | 0.51 (0.39) | 0.11 | 0.58 | 0.35 | 0.62 (0.68) | 0.96 (0.90) | 0.83 (0.81) |
| Average depth available pixels (m) | 66 (74) | 103 (87) | 82 (80) | 131 | 148 | 140 | 67 (67) | 76 (71) | 72 (69) |
| Average depth missing pixels (m) | 67 | 95 | 78 | 131 | 138 | 135 | 27 | 81 | 63 |
| Case 1 pixels/available pixels | 0.63 (0.58) | 0.86 (0.72) | 0.77 0.66 | 0.32 | 0.76 | 0.58 | 0.62 (0.46) | 0.77 (0.65) | 0.71 (0.55) |
| Average depth Case 1 pixels (m) | 80 (86) | 114 (99) | 92 (93) | 143 | 157 | 149 | 81 (80) | 89 (86) | 85 (83) |
| Case 2 pixels/available pixels | 0.14 (0.28) | 0.37 (0.42) | 0.23 (0.34) | 0.24 | 0.68 | 0.42 | 0.23 (0.35) | 0.38 (0.54) | 0.29 (0.45) |
| Average depth Case 2 pixels (m) | 35 (43) | 64 (70) | 46 (55) | 107 | 143 | 128 | 31 (44) | 47 (57) | 38 (52) |
| $Z < Z_{1\%}$ pixels/available pixels | 0.07 | 0.21 | 0.15 | 0.01 | 0.07 | 0.03 | 0.27 | 0.36 | 0.31 |





**Table 4.** Median values of the products used by Gattuso et al. (2006) and the present study.

|  |  | Gattuso et al. (2006) | Present study |
|---|---|---|---|
| Depth (m) | Non polar | 42.5 | 45 |
|  | Arctic | 31.4 | 38.5 |
| $K_{PAR}$ (m$^{-1}$) | Non polar | 0.0968 | 0.1336 |
|  | Arctic | 0.1407 | 0.1630 |
| $PAR$ (mol photons m$^{-2}$ d$^{-1}$) | Non polar | 41 | 41 |
|  | Arctic | 19 | 22 |

**Table 5. Top: Organisms**. Surface area (% of the coastal zone) where irradiance does not limit the distribution of photosynthetic organisms. Values reported by Gattuso et al. (2006) are shown in parentheses for comparative purposes. The irradiance thresholds are the first deciles of the minimum light requirements compiled by Gattuso et al. (2006). Data are not reported in the Arctic region for seagrasses and Scleractinian (reef-building) corals where these groups are not present. **Bottom: Communities**. Surface area (% of the coastal zone) where benthic irradiance is higher that the daily community compensation irradiance (NPP>0). The irradiance thresholds are the first deciles of the minimum light requirements compiled by Gattuso et al. (2006). Data are not reported for seagrass communities and coral reefs in the Arctic and Antarctic regions where they do not occur.

|  |  | Percent surface area in region |  |  |  |  |
|---|---|---|---|---|---|---|
|  | Irradiance | Non polar | Arctic | Antarctic | Total surface area (10$^6$ km$^2$) | |
| **Organisms** |  |  |  |  |  | |
| Seagrasses | 1.3 | 20 (28) | – | – | 3.78 (5.27) | |
| Macroalgae |  |  |  |  |  | |
| – Filamentous and slightly corticated filamentous | 0.2 | 37 (42) | 18 (26) | 4 | 8.21 (9.50) | |
| – Corticated foliose, corticated and foliose | 0.098 | 43 (47) | 23 (30) | 5 | 9.65 (10.68) | |
| – Leathery and articulated calcareous | 0.040 | 50 (54) | 29 (36) | 6 | 11.28 (12.37) | |
| – Crustose | 0.001 | 70 (66) | 49 (51) | 19 | 16.32 (15.55) | |
| Microphytobenthos | 0.4 | 31 (37) | 14 (22) | 3 | 6.73 (8.31) | |
| Scleractinian corals | 0.18 | 38 (43) | – | – | 7.29 (8.09) | |
| **Communities** |  |  |  |  |  | |
| Seagrass beds | 2.4 | 15 (23) | – | – | 2.78 (4.32) | |
| Macroalgal communities | 1.6 | 18 (26) | 8 (13) | 2 | 3.91 (5.71) | |
| Microphytobenthic communities | 0.24 | 36 (41) | 17 (25) | 3 | 7.83 (9.19) | |
| Coral reefs | 4.4 | 10 (19) | – | – | – | – |





## 4  Data availability

1. **The geographical and optical data** generated and used in this paper are openly available at the World Data Center
   PANGAEA: Gentili and Gattuso (2020); https://doi.pangaea.de/10.1594/PANGAEA.910898. It consists of 14 netCDF
   files with an unique dimension (the coastal pixel number) which is identical for all files.

   - a netCDF file with geographical information (latitude, longitude, depth, area) (CoastalLight_geo.nc; about 1.2 Gb)

   - a netCDF file with the climatology over the whole 21 year period calculated as the mean values of the 242 monthly
     data of $PAR$, $K_{PAR}$ and $PAR_B$ (CoastalLight_00.nc; about 1.1 Gb)

   - 12 netCDF file with monthly mean of the 21 monthly values of $PAR$, $K_{PAR}$ and $PAR_B$:

     - Monthly climatology, January: CoastalLight_01.nc (6.2 Gb)

     - Monthly climatology, February: CoastalLight_02.nc (6.8 Gb)

     - Monthly climatology, March: CoastalLight_03.nc (7 Gb)

     - Monthly climatology, April: CoastalLight_04.nc (7 Gb)

     - Monthly climatology, May: CoastalLight_05.nc (7 Gb)

     - Monthly climatology, June: CoastalLight_06.nc (9.6 Gb)

     - Monthly climatology, July: CoastalLight_07.nc (10.6 Gb)

     - Monthly climatology, August: CoastalLight_08.nc (11 Gb)

     - Monthly climatology, September: CoastalLight_09.nc (10.4 Gb)

     - Monthly climatology, October: CoastalLight_10.nc (7.8 Gb)

     - Monthly climatology, November: CoastalLight_11.nc (6.4 Gb)

     - Monthly climatology, December: CoastalLight_12.nc (6 Gb)

2. **The surface area of three regions** (Arctic, Antarctic and non-polar) receiving an irradiance above a certain threshold is
   available using the R package *CoastalLight*: https://github.com/jpgattuso/CoastalLight. To install the package, proceed
   as follows:

   - install.packages("devtools")

   - library(devtools)

   - install_github("jpgattuso/CoastalLight")

3. **The surface area of a subregion** of one of the regions above receiving an irradiance above a certain threshold can be
   derived by (complete information can be found in the documentation of the *CoastalLight* package):

   - connecting to the web server http://obs-vlfr.fr/Pfunction to calculate and download its $\mathcal{P}$-function

   - then using this $\mathcal{P}$-function with function *cl_surface* of the *CoastalLight* package.



## 5  Conclusions

This study builds on the first, and still only, global distribution of photosynthetically available radiations reaching the sea floor (Gattuso et al., 2006). It improves the geographical and depth resolutions, and covers a much longer period of time. Despite these key improvements, several limitations inherent to the approach remain. While the spatial resolution is twice better than the previous products, 4.6 km at the equator it is still coarse for investigating the distribution and function of organisms and communities which change at much finer scales. The parameterization used to convert reflectance data to irradiance is approximate in Case 2 waters. Finally, light absorption in the benthic nepheloid layer is not taken into consideration. The global distribution of $PAR_B$ we provide is derived with state-of-the-art data and computations and is arguably the best that can be offered at this time. Despite its shortcomings, it should considerably improve estimates of the geographical and depth distributions of photosynthetic organisms and ecosystems and help assess their contribution to global biogeochemical cycles.



## Appendix A: Graphical representation of P-functions

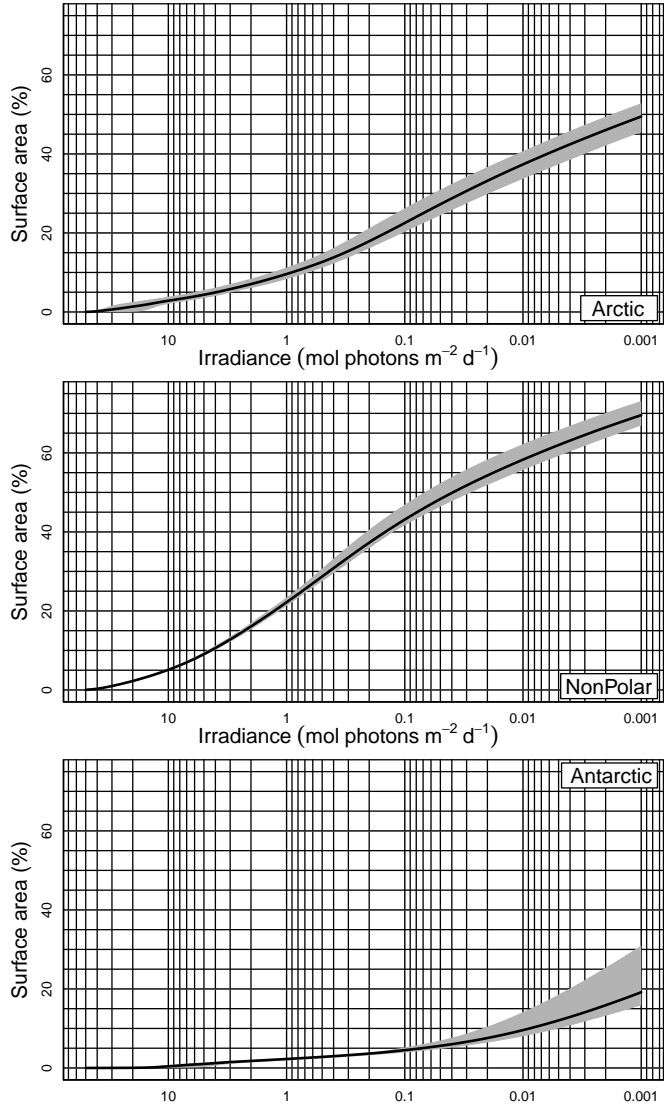

**Figure A1.** Cumulative surface area of the sea floor (S) receiving irradiance above a prescribed threshold ($E_z$). Data are expressed in percent of the total surface area of each region ($19,080,010$, $6,100,532$ and $146,171$ km$^2$, respectively for the non-polar, Arctic and Antarctic regions). The shaded area shows the monthly variability.





*Author contributions.* J.-P.G conceptualized the study. B.G. developed the methods, processed the data, and carried out all data analysis, including software development, with periodic feedback from J.-P.G. J.-P.G wrote the original draft with contributions from B.G., D.A. and D.D. All authors reviewed and edited the final manuscript.

*Competing interests.* None.

5 *Acknowledgements.* This is a product of The Ocean Solutions Initiative (http://bit.ly/2xJ3EV6), with support from Prince Albert II of Monaco Foundation, Veolia Foundation, IAEA Ocean Acidification International Coordination Centre and French Facility for Global Environment; and of the EU-H2020 project INTAROS (grant #727890). This paper is a contribution to an Euromarine project (http://euromarinenetwork.eu/activities/role-macroalgae-global-ocean-carbon-budget) led by Dorte Krause-Jensen. Participants of the foresight workshop held in Granada in 2019, especially Jorge Assis, Carlos Duarte and Dorte Krause-Jensen are gratefully acknowledged for stimulating discussions.





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
