# Peer review of "Global distribution of photosynthetically available radiation on the seafloor"

_Earth System Science Data, 2020_

## Referee Comment (RC1) · Anonymous Referee #1 · 31 Mar 2020

The article, 'Global distribution of photosynthetically available radiation on the seafloor', by Jean-Pierre Gattuso, presents a 21-year time series of benthic PAR. The dataset is an improved version of a prior data set (Gattuso et al., 2006). The current dataset estimates benthic PAR using ocean color and bathymetry data. The time series is four times longer with improved spatial and bathymetric resolution. The article presents a unique dataset, useful for a variety of ecological studies of the benthic community and is therefore of relevance to the scientific community. The time-series does include state-of-the-art ocean color data available to the scientific community. However, the authors are requested to consider the following comments and suggestions.

Major issues

- Depth range in the coastal zone ranges from about 0 – 100/150 m (Fig 4). Considering satellite receives signal only from a top layer of the ocean (referring to the concept of optical depth, the depth from which satellite receives 90% of its signal) and so Kdpar obtained from satellite data represents attenuation from this top layer, kindly explain in the manuscript how PAR obtained using equation 6 is actually bottom PAR. May be provide a schematic to explain the concept.

- A list of symbols and abbreviations used in the article is missing. Add one if possible and maintain consistency with Gattuso et al. 2006 for ease of the reader. For example, Gattuso et al. 2006, used $K_D$ and the present article uses $K_d$

- Page 4, line 25 states spectral composition is not considered in the study. But, throughout the manuscript irradiance is used in place of PAR.

- Page 2, line 2: A number of references (old as well as new) are available that provide relationships between Secchi depth and attenuation.

  https://doi.org/10.1016/S0380-1330(88)71564-6

  https://link.springer.com/article/10.1007/s10750-012-1084-2

  https://doi.org/10.1016/j.rse.2015.08.002

  https://link.springer.com/article/10.1007/s10201-008-0246-4

- Page 3, section 2.1: describe the Globcolor project in a sentence or two for info.

- The article, throughout, refers to the present study as 2019, it needs to change to 2020 or else only stick to 'present study' and avoid mentioning the year.

- Figure panels need to be labelled throughout the manuscript.

- Figure 1 caption: **Availability of remote sensing data (monthly mean) over the 21 years' time-series expressed on percentage**. The other half of the caption regarding surface of the coastal zone is not clear and difficult to understand. Please explain in a different sentence.

- Page 13, Figure 5: P1 P2 P3 not explained in the caption. Y axis refers to irradiance or PAR? Units refer to PAR.

- Page 14, Table 3: $Z_{1\%}$ refers to depth at which benthic irradiance or benthic PAR equals 1% of surface?

- Table 4: Could the increased PAR in the Arctic be attributed to increased sea ice melting? Possible to check and provide evidence if this increase is more prominent in the last decade?

Technical comments

Page 1, line 1: Abstract (delete the period)

Page 1, line 2: global distribution of  (photosynthetically available radiation)

Page 1, line 3: to estimate benthic irradiance or benthic PAR?

Page 1, line 3: avoid using references in the abstract

Page 1, line 16: lowest levels of **food** web

Page 2, line 7: **However, in the coastal ocean, primary production also occurs at the bottom, when enough light reaches the sea floor.**

Page 2, line 10: in the past 10 years. (delete the period)

Page 2, line 14-15:  Irradiance or PAR?

Page 2, line 19: Glud et al. ??

Page 2, line 20: a data layer of benthic irradiance for modelling of species distribution as part of

Page 2, line 31: the characteristics of products used by Gattuso et al. (2006) and **of those** in the present study

Page 3, line 1: Table 1.  characteristics of the products used in Gattuso et al. (2006) and **of those** in the present study.

Page 3, line 21: at level-2 of the processing

Page 4, line 7: It **was** carried out

Page 4, line 13: **(Morel and Belanger, 2006)**

Page 4, line 21: Benthic Irradiance or Benthic PAR?

Page 4, equations 2, 3: explain each of the terms

Page 7, table 2 caption: Values reported in Gattuso et al. (2006) are shown in parentheses for **comparison.**

Page 7, line 14: The surface area of the ocean with **depth** less than 200

Page 7, line 16: **the** Antarctic (60 to 90°S) , respectively covering  24.1, 75.5, and 0.6% **surface area** of the global coastal zone**.**

Page 8, line 4: **In the Arctic and the Antarctic, sunlight is available only during the 5 summer months of the year, i.e., June to October and November to March respectively**.

Page 8, line 5: **Furthermore,** data availability is higher in mid-summer than in ….

Page 8, line 12-13: **In contrast, there is a clear dominance of Case 1 over Case 2 waters (70 vs 30%) in the non-polar region whereas it was more even (55 vs 45%) in Gattuso et al. 2006.**

Page 8, line 15: The present study **uses** remote…

Page 9, line 10: The distribution of $PAR_B$ has changed in the present study compared to Gattuso et al. (2006), …..

Page 9, Figure 2 caption, delete 2019

Page 10, Figure 3: left and the right panel not mentioned in the caption. Y axis in the right panel refers to 2019?  check the axis title

Page 11, Figure 4: left and the right panel not mentioned in the caption

Page 11, line 4:  In the non-polar region, higher the irradiance threshold, larger the difference.

---

## Referee Comment (RC2) · Anonymous Referee #2 · 27 May 2020

essd-2020-33-reviewer recommendations

Overall comments: This manuscript should be accepted for publication pending some editing. The science appears to be sound and results are potentially very useful to a wide range of readers, as the authors note in the Introduction and Conclusions. The role of light in biogeochemical cycles, especially the carbon cycle, is so fundamental that many researchers overlook the important details, such as those presented in this paper. My comments are primarily editorial, with the goal of making the manuscript a bit easier to read. One common challenge for the reader is the authors' frequent use of ambiguous pronouns. For example, starting a sentence wit "It", when the closest singular noun is not what the authors are referring to (e.g., second line of the Abstract and also in the Conclusions). Even more nebulous is beginning paragraphs with "It

is..." when rearranging the topic sentence slightly can provide clarity. Inconsistencies are persistent throughout the manuscript, including in the figures and tables. For example, the authors use non polar, non-polar, Non polar, Non-polar, Non-Polar, and even NonPolar. Many of those usages are highlighted in the manuscript pdf that is annotated with comments (provided). Related to this issue is the placement of "Arctic" and "Non polar" graphs in the figures. In Figure 2, Arctic is on the left, but on the right in figures 3 and 4. Similarly, there is no consistency to heading placement in the tables. Also, in Table 5, please provide units for Irradiance. Are the authors referring to mol photons m-2 d-2 or to percent of surface irradiance. Another ambiguity for the reader is the sparse use of "benthic" when referring to photosynthetic organisms in the "Results and discussion". This ambiguity is particularly problematic when referring to "surface area", which generally appears to refer to surface area of the ocean, though the Figure 3 caption does refer to the "surface area of the sea floor". The authors could revise their wording to clarify for the reader, especially in section 3.4, when they are specifically referring to benthic photosynthesis, productivity, communities, etc.

Please also note the supplement to this comment:
https://www.earth-syst-sci-data-discuss.net/essd-2020-33/essd-2020-33-RC2-supplement.pdf
* * *
[Figure]

**Supplement:**

[revised manuscript text omitted]

---

## Author Comment (AC1) · 23 Jun 2020

We thank both referees for their constructive comments to which we reply below (**RC**: referee comment; **AR**: author reply).

**1   Referee #1**

**RC:** *The article, 'Global distribution of photosynthetically available radiation on the seafloor', by JeanPierre Gattuso, presents a 21-year time series of benthic PAR. The dataset is an improved version of a prior data set (Gattuso et al., 2006). The current*

*dataset estimates benthic PAR using ocean color and bathymetry data. The time se-ries is four times longer with improved spatial and bathymetric resolution. The article presents a unique dataset, useful for a variety of ecological studies of the benthic com-munity and is therefore of relevance to the scientific community. The time-series does include state-of-the-art ocean color data available to the scientific community. However, the authors are requested to consider the following comments and suggestions.*

**AR:** Thank you very much for your constructive comments and suggestions which sig-nificantly improved the manuscript.

1.1 Major issues

**RC:** *Depth range in the coastal zone ranges from about 0 – 100/150 m (Fig 4). Con-sidering satellite receives signal only from a top layer of the ocean (referring to the concept of optical depth, the depth from which satellite receives 90% of its signal) and so Kdpar obtained from satellite data represents attenuation from this top layer, kindly explain in the manuscript how PAR obtained using equation 6 is actually bottom PAR. May be provide a schematic to explain the concept.*

**AR:** The confusion stems from the fact that equation 6 was incomplete. The depth it refers to is the bottom depth. Equation 6 now reads:

$$PAR_B = \exp(-K_{PAR} \times z_B) \tag{1}$$

with $z_B$, bottom depth.

**RC:** *A list of symbols and abbreviations used in the article is missing. Add one if possible and maintain consistency with Gattuso et al. 2006 for ease of the reader. For*

*example, Gattuso et al. 2006, used K D and the present article uses Kd.*

**AR:** All symbols and abbreviations are defined in the text. We therefore do not think that a list of them is needed but are happy to add one if the editor wants. We agree that terms were not used in a consistent manner. We now use $K_{PAR}$ for the attenuation coefficient for $PAR$, as in Gattuso et al. (2006). $K_d$ is the accepted term for diffuse attenuation coefficient for the downward irradiance and a given wavelength.

**RC:** *Page 4, line 25 states spectral composition is not considered in the study. But, throughout the manuscript irradiance is used in place of PAR.*

**AR:** The audience of this manuscript is both optical oceanographers, biogeochemists and ecophysiologists. These communities use different terms for the same quantity. To clarify the matter and avoid any misunderstanding, the following text will be added at the very beginning of the Methods section:

> Irradiance, here downwelling irradiance, can be defined or measured at a specific wavelength or integrated within a specific spectral domain. Photosynthetically Available Radiation ($PAR$ in $\mathrm{mol\ photons\ m^{-2}\ d^{-1}}$) is the amount of light available for photosynthesis, that is in the 400 to 700 nm spectral range. Biogeochemists and ecophysiologists use the term irradiance for the same quantity. Both terms are used synonymously in the present paper.

**RC:** *Page 2, line 2: A number of references (old as well as new) are available that provide relationships between Secchi depth and attenuation.*

https://doi.org/10.1016/S0380-1330(88)71564-6

https://link.springer.com/article/10.1007/s10750-012-1084-2

https://doi.org/10.1016/j.rse.2015.08.002

https://link.springer.com/article/10.1007/s10201-008-0246-4

**AR:** Thank you. A citation of Lee et al. (2015) has been added.

**RC:** *Page 3, section 2.1: describe the Globcolor project in a sentence or two for info.*

**AR:** The following text has been added.

> The GlobColour project generates global ocean colour products by merging data from current and past ocean colour instruments (SeaWiFS, MERIS, MODIS, VIIRS and the two OLCI). Merged products are generated through a weighted average of the level-2 geophysical products (e.g., chlorophyll) from individual missions. The weights are assigned to each mission under the form of a global uncertainty value derived through validation with respect to global databases of field observations. Alternative products are also generated through the Garver-Siegel-Maritorena (GSM) model (Garver and Siegel, 1997, Maritorena et al., 2002, 2010).

**RC:** *The article, throughout, refers to the present study as 2019, it needs to change to 2020 or else only stick to 'present study' and avoid mentioning the year.*
**AR:** Thank you. This mistake is now corrected.

**RC:** *Figure panels need to be labelled throughout the manuscript.*

**AR:** Figure panels are identified in the revised version of the manuscript.

**RC:** *Figure 1 caption: Availability of remote sensing data (monthly mean) over the 21 years' time-series expressed on percentage. The other half of the caption regarding surface of the coastal zone is not clear and difficult to understand. Please explain in a different sentence.*

**AR:** The legend now reads as follows:

> Availability of remote sensing data over the 21-year time-series. Availability is expressed as the monthly mean of the percent area of each latitudinal band covered by the satellite

**RC:** *Page 13, Figure 5: P1 P2 P3 not explained in the caption. Y axis refers to irradiance or PAR? Units refer to PAR.*

**AR:** P1 to P3, which referred to time periods, are not needed and have been removed. Irradiance and $PAR$ are used synonymously. See justification above.

**RC:** *Page 14, Table 3: Z surface 1% refers to depth at which benthic irradiance or benthic PAR equals 1% of surface?*

**AR:** Text changed accordingly.

**RC:** *Table 4: Could the increased PAR in the Arctic be attributed to increased sea ice melting? Possible to check and provide evidence if this increase is more prominent in the last decade?*

**AR:** Changes in the penetration of light in the Arctic Ocean are complex to analyze and predict. The loss of sea ice favours the penetration of light but the increased input of dissolved and particulate matter in the coastal zone (the region of interest in the present paper) resulting from the melting of land ice and permafrost restricts light penetration.

[Figure]

We had a look at data Multisensor Analyzed Sea Ice Extent - Northern Hemisphere (MASIE-NH), Version 1 (downloaded on 19 June from https://nsidc.org/data/G02186/versions/1). There is no obvious change in the daily extent of sea ice (Fig. 1) nor in the average sea ice extent during the period of June to October (the period of interest in the present paper; Fig. 2). A correlation between light penetration and the extent of sea ice may exist at subregional scale but investigating such relationship goes well beyond the scope of this paper. We therefore refrain from making any statement on the role of sea ice loss on benthic $PAR$ in the Arctic.

**1.2 Technical comments**

**RC:** *Page 1, line 1: Abstract (delete the period) Page 1, line 2: global distribution of light (photosynthetically available radiation; PAR)*
**AR:** Is the referee referring to the semi colon? It is justified here.

**RC:** *Page 1, line 3: to estimate benthic irradiance or benthic PAR?*
**AR:** Irradiance and $PAR$ are used synonymously. See justification above.

**RC:** *Page 1, line 3: avoid using references in the abstract*
**AR:** The citation has been removed.

**RC:** *Page 1, line 16: lowest levels of food web*
**AR:** The typo has been corrected, thanks. Web is plural because there are many distinct food webs.

**RC:** *Page 2, line 7: However, in the coastal ocean, primary production also occurs at the bottom, when enough light reaches the sea floor.*

**AR:** The text has been changed accordingly.

**RC:** *Page 2, line 10: in the past 10 years. (delete the period) Page 2, line 14-15: Irradiance or PAR?*
**AR:** Done. Irradiance and $PAR$ are used synonymously. See justification above.

**RC:** *Page 2, line 19: Glud et al. ??*
**AR:** Missing year added.

**RC:** *Page 2, line 20: a data layer of benthic irradiance for modelling of species distribution as part of*
**AR:** The text has been changed accordingly.

**RC:** *Page 2, line 31: the characteristics of products used by Gattuso et al. (2006) and of those in the present study*
**AR:** The text has been changed accordingly.

**RC:** *Page 3, line 1: Table 1. Main characteristics of the products used in Gattuso et al. (2006) and of those in the present study.*
**AR:** The text has been changed accordingly.

**RC:** *Page 3, line 21: at level-2 of the processing Page 4, line 7: It was carried out Page 4, line 13: (Morel and Belanger, 2006) Page 4, line 21: Benthic Irradiance or Benthic PAR?*
**AR:** Correction made. Irradiance and $PAR$ are used synonymously. See justification above.

**RC:** *Page 4, equations 2, 3: explain each of the terms Page 7, table 2 caption: Values reported in Gattuso et al. (2006) are shown in parentheses for comparison.*
**AR:** Every term is now defined. Change in the caption done.

**RC:** *Page 7, line 14: The surface area of the ocean with depth less than 200*
**AR:** Done.

**RC:** *Page 7, line 16: the Antarctic (60 to 90S) regions, respectively covering 24.1, 75.5, and 0.6% surface area of the global coastal zone.*
**AR:** Done.

**RC:** *Page 8, line 4: In the Arctic and the Antarctic, sunlight is available only during the 5 summer months of the year, i.e., June to October and November to March respectively.*
**AR:** Done.

**RC:** *Page 8, line 5: Furthermore, data availability is higher in mid-summer than in . . . .*
**AR:** Done.

**RC:** *Page 8, line 12-13: In contrast, there is a clear dominance of Case 1 over Case 2 waters (70 vs 30%) in the non-polar region whereas it was more even (55 vs 45%) in Gattuso et al. 2006.*
**AR:** Done.

**RC:** *Page 8, line 15: The present study uses remote. . .*
**AR:** Done.
**RC:** *Page 9, line 10: The distribution of PAR B has changed in the present study compared to Gattuso et al. (2006), …...*
**AR:** Done.

**RC:** *Page 9, Figure 2 caption, delete 2019*
**AR:** Done.

**RC:** *Page 10, Figure 3: left and the right panel not mentioned in the caption. Y axis in the right panel refers to 2019? check the axis title*
**AR:** The figure and legend have been corrected accordingly.

**RC:** *Page 11, Figure 4: left and the right panel not mentioned in the caption*
**AR:** Now they are.

**RC:** *Page 11, line 4: As shown in fig. 3, In the non-polar region, higher the irradiance threshold, larger the difference.*
**AR:** Text modified accordingly.

**2  Referee #2**

**RC:** *This manuscript should be accepted for publication pending some editing. The science appears to be sound and results are potentially very useful to a wide range of readers, as the authors note in the Introduction and Conclusions. The role of light in biogeochemical cycles, especially the carbon cycle, is so fundamental that many researchers overlook the important details, such as those presented in this paper. My*

*comments are primarily editorial, with the goal of making the manuscript a bit easier to read.*
**AR:** Thank you very much for your constructive comments and suggestions.

**RC:** *One common challenge for the reader is the authors' frequent use of ambiguous pronouns. For example, starting a sentence wit"I", when the closest singular noun is not what the authors are referring to (e.g., second line of the Abstract and also in the Conclusions). Even more nebulous is beginning paragraphs with "It is. . ." when rearranging the topic sentence slightly can provide clarity.*
**AR:** We believe this issue has been addressed in the revised version of the manuscript.

**RC3:** *Inconsistencies are persistent throughout the manuscript, including in the figures and tables. For example, the authors use non polar, non-polar, Non polar, Non-polar, Non-Polar, and even NonPolar. Many of those usages are highlighted in the manuscript pdf that is annotated with comments (provided).*
**AR:** We agree and now use nonpolar, which is a correct English term, throughout the manuscript.

**RC:** *Related to this issue is the placement of "Arctic" and "Non pola" graphs in the figures. In Figure 2, Arctic is on the left, but on the right in figures 3 and 4. Similarly, there is no consistency to heading placement in the tables. Also, in Table 5, please provide units for Irradiance. Are the authors referring to mol photons m-2 d-2 or to percent of surface irradiance.*
**AR:** We agree. Now the regions are shown in the same order in all tables and figures: Arctic, nonpolar and Antarctic. The unit of irradiance is now provided in Table 5.

**RC:** *Another ambiguity for the reader is the sparse use of "benthic" when referring to*

[Figure]

*photosynthetic organisms in the "Results and discussio". This ambiguity is particularly problematic when referring to "surface area", which generally appears to refer to surface area of the ocean, though the Figure 3 caption does refer to the "surface area of the sea floo". The authors could revise their wording to clarify for the reader, especially in section 3.4, when they are specifically referring to benthic photosynthesis, productivity, communities, etc.*

**AR:** Comment addressed in the revised version of the manuscript.

―――――――――――――――――――――

**NSIDC/NIC Sea Ice Product G02186**

Daily Ice Extent

**Fig. 1.** Daily values of sea ice extent in the Northern hemisphere.

**NSIDC/NIC Sea Ice Product G02186**

Mean sea ice extent from June to October

Fig. 2. Monthly means of sea ice extent in the Northern hemisphere.